# Time Series Prediction of Dam Deformation Using a Hybrid STL–CNN–GRU Model Based on Sparrow Search Algorithm Optimization

Chuan Lin [1], Kailiang Weng [1,2,*], Youlong Lin [3], Ting Zhang [1], Qiang He [2] and Yan Su [1]

1   College of Civil Engineering, Fuzhou University, Fuzhou 350108, China
2   Fujian Provincial Investigation, Design & Research Institute of Water Conservancy & Hydropower Co., Ltd., Fuzhou 350001, China
3   Fujian Xiyuan Reservoir Management Office, Fuzhou 350108, China
*   Correspondence: wkl961029@126.com

**Abstract:** During its long service life, an arch dam affected by a combination of factors exhibits a typical time-varying characteristic in terms of its structure and material properties, and the deformation in the dam structure can directly and reliably reflect the health and service status of dams. Therefore, an accurate deformation prediction is an important part of dam safety monitoring. However, due to multiple factors, dam deformation data often tend to be highly volatile, and most existing deformation estimation techniques employ a single algorithm, which may not effectively capture the potential change process. A hybrid model for dam deformation prediction has been proposed to overcome this problem. First, dam deformation data are decomposed into three components by seasonal and trend decomposition using loess. Second, a convolutional neural network–gated recurrent unit (GRU) hybrid model, which optimizes hyperparameters using the sparrow search algorithm, is used to capture the nonlinear relationships that exist in each component. Finally, the final prediction result of dam deformation is the comprehensive output of multiple submodules. The deformation monitoring data (period: 2009–2019) of a parabolic variable-thickness double-curved arch dam located in China are considered as the survey target. The test results indicate that the proposed model is suitable for short-term and long-term prediction and outperforms other models in terms of higher robustness to abnormal sequences than other conventional models ($R^2$ differs by 5.50% and 7.87%, respectively, in short-term and long-term predictions for different measurement points, while other models differ by 9.78% to reach 15.71%, respectively). Among the models studied, the GRU shows better robustness to abnormal series than the LSTM with good prediction accuracy, fewer parameters, and a simpler structure. Hence, the GRU can be employed for dam deformation prediction in practical engineering.

**Keywords:** dam deformation; decomposition and ensemble; deep learning technique; gated recurrent unit

## 1. Introduction

Dams have a long history in China as a comprehensive utilization project for power generation, flood control, irrigation, and water supply. Nowadays, because of various reasons, the risk of dam failure is increasing. For instance, extreme weather can increase the load acting on the dam, and material aging reduces durability [1]. Various monitoring equipment are typically installed in a dam to monitor its structural health at any time, to assess the multidimensional safety status of the dam, including environmental variables. Among the various structural responses of the dam, deformation is relatively simple to measure and can reliably reflect the overall service status of the dam [2]. The purpose of dam deformation monitoring is to maximize the use of the effective information of observation, so as to build a hazard warning model; with this model, the predicted value can be compared with the actual deformation to evaluate the changes in the dam performance.

When the deviation exceeds a preset threshold, a risk alert is raised to identify the dangerous structure response promptly and avoid or reduce its impact [3]. Therefore, improving the accuracy of the dam deformation prediction model has important research significance in the field of dam safety monitoring.

Dam deformation is affected by many factors, such as hydrological conditions around the project and its own material properties [4]. Therefore, it often presents the characteristics of diversity and time variation under the comprehensive influence of multiple factors [5]. In addition, the series of deformation data obtained during the process of construction and use of the dam tend to be highly volatile because of environmental factors and the performance of the collection equipment, including force majeure due to lightning strikes, heavy rains, and network environments. In short, the complexity of the dam deformation evolution law not only limits the accuracy of the prediction model to a certain extent but also increases the difficulty of building prediction models [6]. In the past few decades, many predictive models that can be roughly divided into three categories have been developed, namely physical models, conceptually lumped models, and data-driven models [7]. Generally, physical models help understand the process of dam deformation; however, building such models often requires a large-scale site, equipment, materials, and manpower. The conceptually lumped model involves constructing a numerical prediction model based on the dam structure and physical mechanism. However, the assumption and simplification of boundary conditions in the actual implementation process can avoid the complicated model building process but reduce the accuracy of model prediction, so making it unsuitable to accurately express the nonlinear characteristics of dam deformation [8]. Data-driven models typically use statistical models that involve hydrostatic-season-time (HST) modeling and hydrostatic-seasonal-time thermal (HSTT) modeling. They yield satisfactory prediction results without having to know the detailed mechanical information related to observation data, with specific physical explanations, high calculation speed, and a simple structure [9]. For dam deformation, conventional statistical models have been used to capture the complicated nonlinear characteristics in the past; yet, most statistical models define that the dam deformation is linear, which reduces their reliability and accuracy to a certain extent. Recently, owing to the gradual maturity of artificial intelligence, many data-driven artificial intelligence models have been extended to various complex dam monitoring problems, such as gaussian process regression (GPR) [10], tree-based ensemble models [11], the convolutional neural network (CNN) [12], and the long short-term memory (LSTM) network [13].

Fundamentally, the calculation process of an artificial intelligence model involves mapping some input variables to output variables. Due to the dependence of the time series, the sliding window method is typically used to reconstruct the data. To realize the process of mapping the previous time step tensor to the next required prediction time step, they are, respectively, provided to the nonlinear artificial intelligence algorithm for training. Therefore, the purpose is to explore the nonlinear relationship between the two tensors, and the reconstructed time series prediction is transformed into an artificial intelligence learning problem.

For example, if a 30-day dam deformation dataset is used to train a model, and the purpose is to predict dam deformation the next day, which is called the prediction time, the time step needs to be first determined to generate samples. If the time step is five days, the first sample for the model training takes the dam deformation data from the first day to the fifth day as the input and the dam deformation data of the sixth day as the output. The next sample is inputted with the dam deformation data from the second day to the sixth day and outputs the dam deformation data from the seventh day. The sample generation process continues until the time corresponding to the output variable generated by the model is the 30th day. The prediction time, which is one day in this case, is determined by the actual monitoring requirements. The time step size should be chosen depending on the characteristics of the data, and it is difficult to determine its value without any prior knowledge. Therefore, before using the artificial intelligence model to

predict dam deformation, it is necessary to select the best time step in the model to capture the dependence between the original time series and improve the prediction accuracy.

As one of the data-driven artificial intelligence models, the LSTM relies on the advantages of resisting noise in the time series without prior assumptions and is applied to solve nonlinear, high-dimensional, and large-sample problems [14]. Moreover, the LSTM has been widely used in the analyses of dam deformation and machinery. For example, Zhang et al. [15] proposed an improved LSTM model, which divides a variable into delay variables and no delay variables, for dam displacement prediction. Liu et al. [16] obtained the residual stiffness related to the fatigue life of a blade under fatigue test conditions based on the LSTM and combined with historical fatigue data. However, the performance of the LSTM is significantly affected by the choice of the computational parameters employed in practice, such as the dropout rate used to avoid overfitting and the number of cell units in the hiding layers. However, there is no consensus on the optimal value selection. Furthermore, for a higher-precision dam deformation response, a high computational memory is required for training the model, making it challenging to use data-based sophisticated models for dam deformation prediction. Recently, as a variant of the LSTM, the gated recurrent unit (GRU) [17] has shown comparable performance to LSTM in many research fields, with its simpler structure and higher calculation speed [18]. However, the GRU network has drawbacks, such as overfitting and parameter tuning, which limit its application. In addition, it is more difficult to construct a model using a single algorithm because of the noise and heteroscedastic structure of time series data [19]. Combinatorial models have been developed to overcome this problem; the model built for predicting the target variable is sophisticated and is no longer a single-algorithm model but a diversified complex network. It makes full use of the effective information in each model by connecting each model with the weights to improve the model accuracy [20]. For instance, Kanjo et al. [21] applied a hybrid model (CNN–LSTM) to capture the diversity of multimodal data at the sensor and feature level, which is conducive to emotion classification modeling. Huang et al. [22] verified the necessity of fully extracting sequence features when the CNN–LSTM model is used to predict PM2.5 concentration. These hybrid model-related studies demonstrated that owing to the extraction potential of the representative features of the CNNs and effective gated structure of multilayered LSTM or GRU, the model can more strongly express the temporal features in terms of the computational complexity and prediction accuracy, with the lowest error rate.

Due to the particularity of the dam deformation monitoring data, although the commonly used dam deformation prediction methods can provide good prediction results, their comprehensive prediction performance is poor in the face of complex original observation datasets. The robustness of these methods is poor, and the long-term forecast will accumulate errors. Table 1 is a comparison table of the advantages and disadvantages of common dam deformation prediction methods.

Notably, because of the many factors affecting dam deformation, existing methods have limited capabilities in modeling the multivariate correlation and high volatility of dam deformation, and to achieve the required accuracy, the final model architecture becomes complex and less universal. Therefore, the main method currently used is to strip out representative subcomponents from the observed dam deformation time series to reduce the modeling difficulty. For example, Rehman et al. [23] used multiple real-valued projections of signal to improve empirical mode decomposition (EMD) for the problem that it is difficult to find local extremum of the multivariate signal, and then remove noise from observations on dam safety. Bruce et al. [24] used wavelet analysis (WA) to decompose the localized waveforms of dam displacement time series for model construction. However, these methods have strict requirements on the assumption of the original sequence, and the denoising effect of white noise is poor, thereby decreasing the ability to reproduce abnormal values in certain tasks, and the application scope is not wide. The seasonal-trend decomposition based on loess (STL) is a mature filtering program that can split the original time series into several representative variation components for subsequent analysis. Owing to

its strong reproducibility for outliers in the time series and its ability to generate robust subseries, the STL is recognized in many engineering fields involving dam deformation decomposition [25]. On the other hand, to deal with the problem of artificial intelligence parameter tuning, the sparrow search algorithm (SSA) has recently been proposed to deal with the parameter global optimization problem due to its advantages of simplicity of implementation, ease of expansion, and self-organization [26]. This paper proposes a hybrid model for dam deformation prediction and uses the SSA to perform global optimization of the model parameters. Taking a real-world parabolic variable-thickness double-curved arch dam in China as an example, the feasibility of the model is successfully verified. The main contributions of this paper can be summarized as follows: (i) The STL is used to decompose the complex dam deformation time series into several subcomponents, which makes data features in the dam deformation series more prominent and reduces the difficulty of subsequent modelling. (ii) For each subcomponent, the CNN–GRU model is used to identify the mapping relationship between impact factors and target variables, which can make the whole prediction process structured and broaden the application range of the model. (iii) The SSA method is used to identify the appropriate combination of the parameters to endow the model with optimization capability, accomplishing the pertinent modelling of dam deformation series at different measurement points. (iv) Compared with several conventional models, the proposed hybrid model provides better prediction accuracy, long-term stability, and robustness to abnormal sequences.

**Table 1.** Comparison of advantages and disadvantages of dam deformation prediction methods.

| Method Performance | Model Analysis Method | Process Line Analysis Method | Intelligent Analysis Method | Intelligent Combinatorial Models |
|---|---|---|---|---|
| Advantage | Clear physical and mechanical relationship and dynamic criterion | Simple prediction principle and easy operation | Avoid physical and mechanical relationship analysis, intelligent prediction | Provide good prediction results for complex situations |
| Disadvantage | Model performance is sensitive and practicality is limited | Huge subjective influence and static representation analysis | Insensitive to the noise and heteroscedastic structure of time series data | High calculation cost |
| Status | The existing methods for dam deformation prediction still exist and actual engineering problems need to be solved urgently, such as sensitive to monitoring data size, insensitive to the noise and heteroscedastic structure of time series data, and low efficiency in calculation and so on. | | | |

The remainder of this paper is organized as follows. Section 2 presents a brief background on STL, CNN, LSTM, GRU, and SSA, and describes the construction process of the proposed hybrid prediction model for dam deformation. Moreover, we introduce a case study and experimental plan, simulation and analysis results of the proposed model, and a comparison with other models in Section 3. Finally, conclusions and future work are summarized in Section 4.

## 2. Methods

### 2.1. Seasonal-Trend Decomposition Procedure Based on Loess

The STL is a versatile and robust method of time series decomposition, it assumes that the time series is an additive model composed of trends, seasons, and remaining components [27]. The STL is usually used for time series with great volatility and instability due to it is convenience to calculate and not being sensitive to abnormal signals in the data. The components with independent characterization characteristics can be decomposed through this model to facilitate subsequent analysis [28]. Using this model, the dam deformation time series ($Y_t$) results from the summation of three components:

$$Y_t = S_t + T_t + R_t (t = 1, \ldots, T) \tag{1}$$

where $S_t$ is the seasonality, $T_t$ is the trend, and $R_t$ is the residual.

The STL decomposition method needs to define decomposition frequency parameters for the seasonal component. It was set at 365 due to the dam deformation having annual periodicity.

### 2.2. Convolutional Neural Networks

A convolutional neural network (CNN) is a feedforward neural network that shares weights among layers, and is considered a simple model with powerful representation capabilities [29]. They were originally used for two-dimensional signals such as voice and images, commonly called "2D-CNNs". Two-dimensional CNNs extract features from a complex and large number of multidimensional tensors and capture the law of changes. In order to enable classic CNNs to meet more engineering needs, Kiranyaz et al. proposed the first 1D-CNN that can integrate feature extraction and classification into a unified network [30]. Since then, 1D-CNNs have gradually been applied in various research fields, and many breakthrough results have been achieved.

As stated before, 1D-CNNs are used in this work for feature extraction, the structure of 1D-CNNs is shown in Figure 1. The dam deformation time series have fixed period. Therefore, the 1-dimensional convolution (1D Conv) layer is equivalent to a signal filter that extracts the features of original time series. Typically, a sigmoid activation function is used for feature classification. Pooling layers are usually used to compress the amount of data and parameters to reduce overfitting in CNN models. In our proposed model, a maximum pooling layer is used, and the maximum value in the field of view of the convolution kernel in the previous convolution layer is retained to achieve the purpose of feature dimensionality reduction.

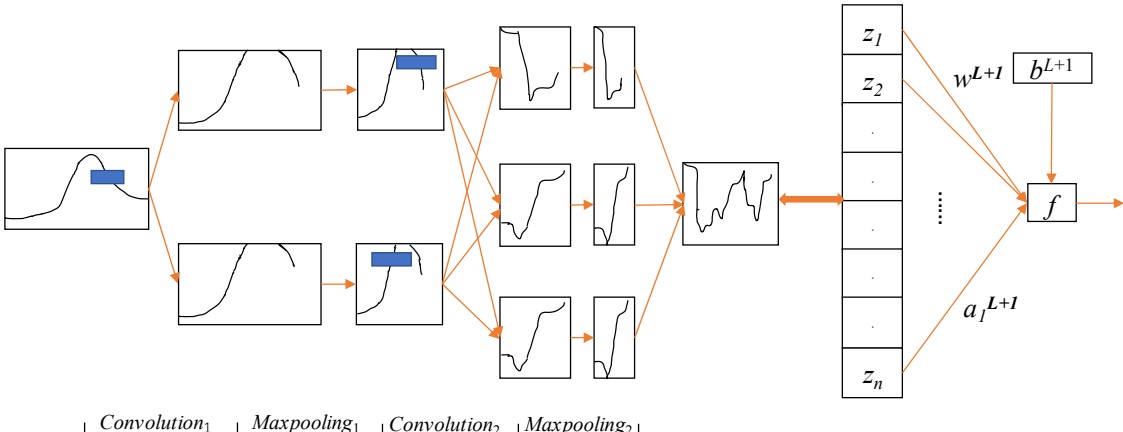

**Figure 1.** One-dimensional convolutional neural network architecture.

### 2.3. Gated Recurrent Neural Network

#### 2.3.1. Long Short-Term Memory Unit

Recurrent neural network (RNN) has units to memorize long-term dependencies and was initially proposed for sequence data. However, with time lags increasing, gradients of RNNs may vanish through unfolding RNNs into deep forward neural networks. In order to solve this problem, the long short-term memory (LSTM) unit was initially proposed by Hochreiter and Schmidhuber, which has a similar structure and better sequence processing capabilities to traditional RNN [31]. The form of LSTM is shown in Figure 2, where $x_t$ is the input matrix, $h_t$ is the output of the hidden layer in the network at time step $t$, $c_t$ is the cell state of the memory cell at time step $t$. For the next time step, the LSTM cell accepts the cell state ($c_t$) and hidden state ($h_t$) of the previous time step. To better learn long short-term dependencies of sequence data, three special structures called gates are introduced in LSTM network. The gates are input gate ($i_t$), forget gate ($f_t$), and output gate ($o_t$). These gates act as filters, serving different purposes.

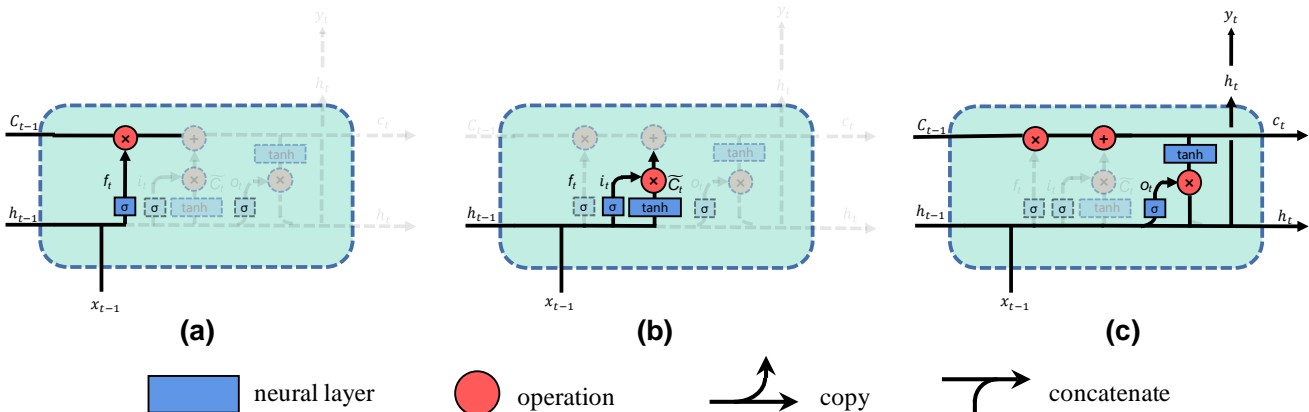

**Figure 2.** The LSTM architecture: (**a**) the operating mechanism of forget gate; (**b**) the operating mechanism of input gate; and (**c**) the operating mechanism of output gate.

Input gate ($i_t$) takes the new vector and combines it with the existing vector to create an updated value. Whether it is necessary to input new data is determined by the sigmoid layer of the input gate layer. The existence of forget gate ($f_t$) enables memory cells to have the ability to delete attribute information, thereby storing information that can better characterize time lag. Output gate ($o_t$) determines output value of LSTM model according to the cell state. Thanks to the design of these gates, LSTM model has a significant improvement in capturing the long- and short-term dependencies between sequences compared with the traditional RNN [32]. In dam deformation predicted model, a rectified linear unit (ReLU) layer is applied on output layer of LSTM model. Moreover, the recurrent dropout mechanism is designed in LSTM model to avoid overfitting during the training process.

In Figure 2, the cell state ($c_t$) and hidden state ($h_t$) of the LSTM cell are calculated as follows:

$$f_t = \sigma(W_{xf}x_t + W_{hf}h_{t-1} + b_f) \tag{2}$$

$$i_t = \sigma(W_{xi}x_t + W_{hi}h_{t-1} + b_i) \tag{3}$$

$$\hat{c}_t = \tanh(W_{xc}x_t + W_{hc}h_{t-1} + b_c) \tag{4}$$

$$c_t = f_t \otimes c_{t-1} + i_t \otimes \hat{c}_t \tag{5}$$

$$o_t = \sigma(W_{xo}x_t + W_{ho}h_{t-1} + b_o) \tag{6}$$

$$\hat{c}_t = \tanh(W_{xc}x_t + W_{hc}h_{t-1} + b_c) \tag{7}$$

where $\sigma$ is the logistic sigmoidal function, $\otimes$ is element-wise multiplication of two vectors, and $W_{xi}$, $W_{hi}$, $W_{xf}$, $W_{hf}$, $W_{xo}$, $W_{ho}$, $W_{xc}$, and $W_{hc}$ are the network weights matrices. Similarly, $b_i$, $b_f$, $b_o$, and $b_c$ are bias vectors. $f_t$, $i_t$, and $o_t$ are vectors for the activation values of the forget gate, the input gate, and the output gate, respectively.

### 2.3.2. Gated Recurrent Unit

LSTM is known for its excellent ability to capture the long- and short-term dependence of sequence data. However, the tensor calculation of LSTM networks is complicated, which will cause the training process to often take a long time. To reduce the training time, gated recurrent unit (GRU) networks improve the inherent gate structure of LSTM network, and shows better results on certain tasks [17].

Similar to the LSTM, GRU also controls the information exchange of hidden cells through gate units; however, the difference is that GRU does not have a separate memory cell. The detailed implementation of GRU is shown in Figure 3. The difference from LSTM is that GRU combines the hidden state ($h_t$) and the cell state ($c_t$) into one. Therefore, there are only two special gate structures left in the GRU, namely the update gate ($z_t$) and the reset gate ($r_t$). The update gate ($z_t$) determines the degree to which the hidden layer state

information in the previous time step $t - 1$ is accepted by the hidden layer in the current time step $t$. Reset gate ($r_t$) determines the extent to which the hidden layer state information from the previous moment is recorded in the candidate hidden layer state $\hat{c}_t$ at the current moment. The values of both the update gate and the reset gate are directly proportional to the amount of state information introduced in the previous time step.

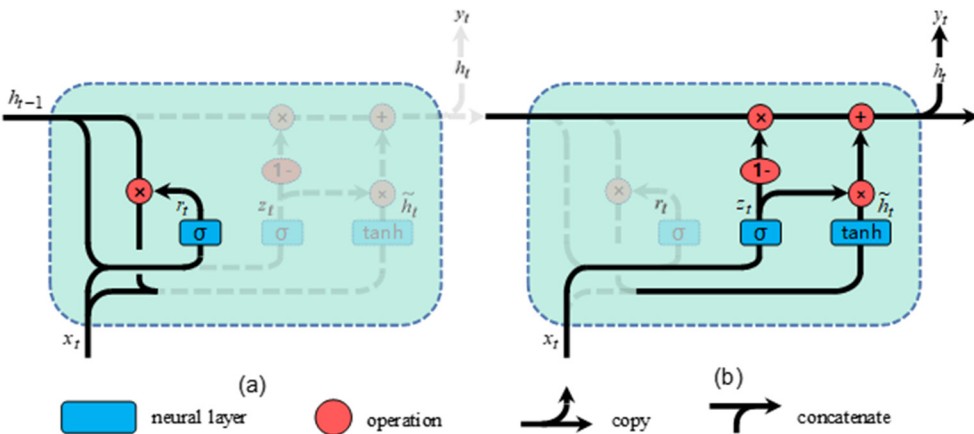

**Figure 3.** The GRU architecture: (**a**) the operating mechanism of reset gate; (**b**) the operating mechanism of update gate.

In Figure 3, the GRU networks are calculated as follows:

$$r_t = \sigma(W_{xr}x_t + W_{hr}h_{t-1} + b_r) \tag{8}$$

$$z_t = \sigma(W_{xz}x_t + W_{hz}h_{t-1} + b_z) \tag{9}$$

$$\hat{c}_t = \tanh(W_{xc}x_t + W_{hc}(r_t \otimes h_{t-1}) + b_c) \tag{10}$$

$$c_t = (1 - z_t) \otimes c_{t-1} + z_t \otimes \hat{c}_t \tag{11}$$

$$h_t = c_t \tag{12}$$

where $W_{xr}$, $W_{hr}$, $W_{xz}$, and $W_{hz}$ are the network weight matrices. $b_r$ and $b_s$ are bias vectors, and $r_t$ and $z_t$ are vectors for the activation values of the update gate and reset gate, respectively.

### 2.4. Sparrow Search Algorithm

The sparrow search algorithm (SSA) is a metaheuristic algorithm, and its operation process belongs to the group intelligence optimization based on the socialized feature optimization of the group, which has stronger parameter search ability and faster efficiency [33]. It is inspired by the foraging behavior and anti-predatory behavior of sparrows, and the specific bionic principles are as follows:

(1) Total sparrow populations can usually be abstracted into explorer and follower models. Explorers mainly provide foraging directions and search areas for the whole population, while followers follow the explorers in order to enhance model adaptation.

(2) Sparrows usually engage in immediate anti-predatory behavior when the entire population is threatened by a predator or when they are aware of danger. Therefore, some of the sparrows will be selected as vigilantes during the food search process to cooperate with the population to keep away from predators.

(3) In the process of searching for food, explorers and followers can dynamically switch with each other to obtain a better foraging decision.

(4) In order to increase the predation rate of the population, some of the followers will monitor the explorers to obtain better foraging directions and search areas to obtain more food.

Therefore, the specific operation flow of SSA can be summarized as follows, and its flow chart is shown in Figure 4.

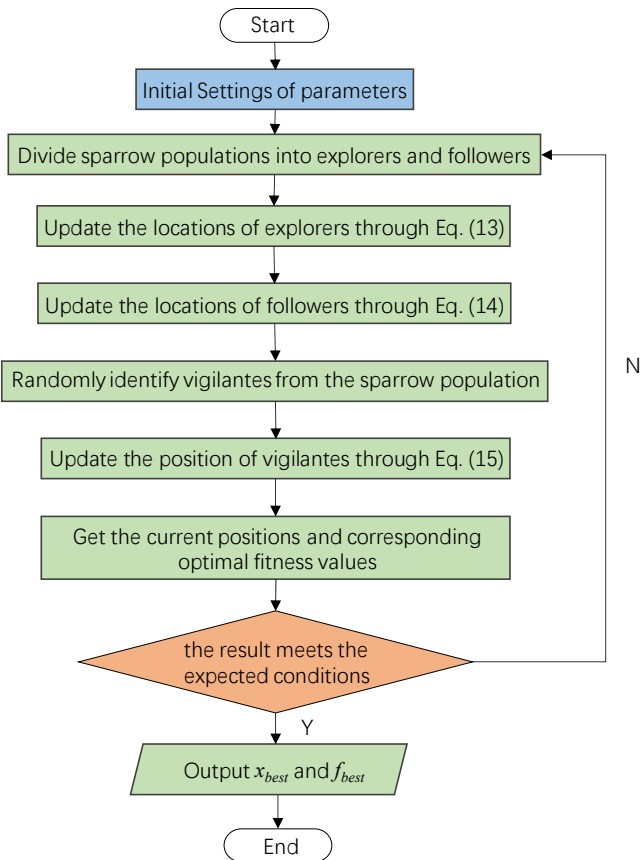

**Figure 4.** Framework of the SSA optimization procedure.

**Step 1** Initial settings are made for the algorithm parameters, including the population size of sparrows *p_size*, the number of explorers *en*, the number of followers *fn*, the number of vigilantes *vn*, the maximum iterations $g_{max}$, the dimension of the search space *D*, and the upper limit *ub* and lower limit *lb* of the search domain. It can be assumed that in a *D*-dimensional search space, there exist *N* sparrows, then the position of the *i*-th sparrow in the *D*-dimensional search space is $X_i = [x_{i1}, \cdots, x_{id}, \cdots, x_{iD}](i = 1, 2, \ldots, p\_size)$ where $x_{id}$ denotes the position of the *i*-th sparrow in the *d*-th dimension.

**Step 2** Update the explorer's position with the relevant calculation equation as shown in Equation (13).

$$x_{id}^{t+1} = \begin{cases} x_{id}^t \cdot \exp(\frac{-i}{\alpha T}) & R_2 < S_T \\ x_{id}^t + QL & R_2 \geq S_T \end{cases} \tag{13}$$

where *t* is the current number of iterations, *T* is the maximum number of iterations, *α* is a uniform random number between (0, 1], *Q* is a random number obeying the standard normal distribution, *L* is a matrix of size $1 \times d$ and all elements are 1, $R_2 \in [0, 1]$ and $S_t \in [0.5, 1]$ are the warning and safety values, respectively. When $R_2 < S_T$, the population does not detect the presence of predators or other dangers, the foraging environment is safe, and the explorer can conduct an extensive search to guide the population to a higher degree of adaptation; when $R_2 \geq S_T$, the vigilantes detect the danger and immediately release the danger signal, and the population immediately engages in anti-predatory behavior, adjusts its search strategy, and quickly proceeds to a safe area.

**Step 3** Update the position of the follower according to Equation (14).

$$x_{id}^{t+1} = \begin{cases} Q \cdot \exp(\frac{xw_d^t - x_{id}^t}{i^2}) & i > \frac{n}{2} \\ xb_d^{t+1} + |x_{id}^t - xb_d^{t+1}| A^+ \cdot L & else \end{cases} \tag{14}$$

where $A$ is a matrix of $1 \times D$ dimensions, $xw_d^t$ is the most unfavorable position of the sparrow in the $d$-th dimension at the $t$-th iteration of the population, and $xb_d^{t+1}$ is the optimal position of the sparrow in the $d$-th dimension at the $t + 1$-th iteration of the population. When $i > n/2$, it indicates that the $i$-th follower does not get food, is in a hungry state, has a low adaptation level, and needs to fly to other places for foraging in order to obtain higher energy; conversely, the $i$-th follower will find a random position near the current optimal position $xb$ for foraging.

**Step 4** To ensure effective anti-predatory behavior, 10–20% of the sparrows in the population are randomly identified as vigilantes, and their positions are updated according to Equation (15).

$$x_{id}^{t+1} = \begin{cases} xb_d^t + \beta(x_{id}^t - xb_d^t) & f_i \neq f_g \\ x_{id}^t + K(\frac{x_{id}^t - xw_d^t}{|f_i - f_w| + e}) & f_i = f_g \end{cases} \tag{15}$$

where $\beta$ is the step control parameter and is a normally distributed random number obeying a mean of zero and a variance of one; $K$ is a random number between $[-1, 1]$, which indicates the direction of sparrow movement and is also the step control parameter; $e$ is a very small constant to avoid a denominator of zero; $f_i$ is the fitness value of the $i$-th sparrow; $f_g$ and $f_w$ are the optimal and worst fitness values of the current sparrow population, respectively. When $f_i \neq f_g$, it means that the sparrow is at the edge of the population and is vulnerable to predator attack; when $f_i = f_g$, it means that the sparrow is in the middle of the population and is aware of the threat of predators and adjusts its search strategy by approaching other sparrows in time to avoid being attacked by predators.

**Step 5** Obtain the current positions of all species of sparrows and update their corresponding optimal fitness values.

**Step 6** Steps 2 to 5 are repeated until the result reaches the maximum number of iterations $g_{max}$ or meets the expected error, and the best position $x_{best}$ of each type of sparrow and its corresponding best fitness value $f_{best}$ are output.

*2.5. Proposed Method for Dam Deformation Time Series Prediction*

To improve the predicted accuracy of dam deformation prediction, we propose the hybrid model as shown in Figure 5. This model first uses STL to decompose observed dam deformation data into different representative subcomponents. Second, the SSA method optimizes the hyperparameters in the architecture of CNN and GRU. Finally, the output results of all subcomponents predicted by CNN–GRU are added, and the final prediction result of the dam deformation at the future time is obtained. The proposed method is mainly divided into two stages: decomposition and ensemble. In the decomposition stage, the model uses STL to decompose observed dam deformation data into multiple simple subcomponents, thereby reducing the difficulty of subsequent modeling. In the ensemble stage, the feature information of the subseries is extracted and merged to obtain the final results. This hybrid method can not only reduce the volatility of the original time series but also improve the generalization ability of the forecasting model.

The proposed model runs according to the following steps.

Step 1: **Data acquisition.** The corresponding dam monitoring data are acquired through the sensors installed in the dam. The HST model focuses on the correlation between water depth, temperature, and dam deformation for calculation reduction to determine the key factors that affect the performance of dams.

The key factor datasets are normalized to range of [0, 1] and divided into training set and testing set as:

$$x_i' = \frac{x_i - \mu}{\sigma} \tag{16}$$

where $x_i'$ and $x_i$ are the normalized and original value of the $i$-th data sample, respectively. $\mu$ is the mean value of training data and $\sigma$ is the standard deviation of training data.

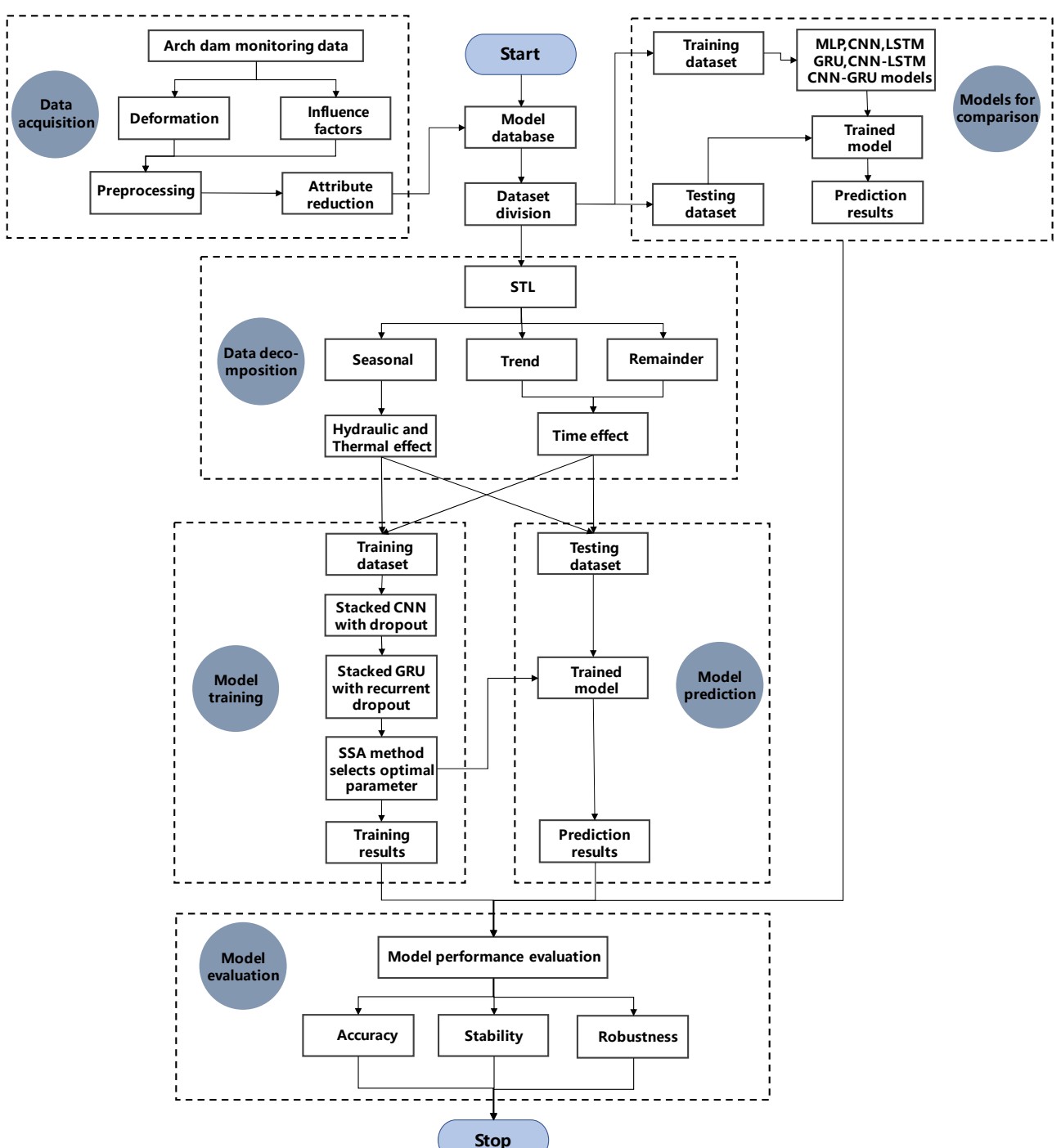

**Figure 5.** Sketch map of the proposed model for dam deformation prediction.

Step 2: **Data decomposition.** Using STL to extract representative sequences of original dam deformation time series, including seasonal $S_t$, trend $T_t$, and remainder $R_t$. The seasonal component is the periodic characteristic response of the water level and seasonal temperature received during the operating period of the dam. The trend component and the remainder component comprehensively reflect the degradation process of concrete material properties and the compression deformation law of geological structures. The dataset is divided with the corresponding influencing factors.

Step 3: **Model training.** The outliers removed and normalized sequence are used as input samples for model training. In training process, the gradient descent algorithm is used to back propagation error to optimize state of each hidden layer. Using the dropout

algorithm to overcome the potential overfitting problem during training. For the model optimization, primarily, it needs to declare the search space. Second, the fitness function is determined for *RMSE*. When its value is small, it means that it has higher fitness, and is the better prediction performance of model. Third, the optimization classifier is built. The SSA method searches the optimal parameters for all CNN–GRUs, such as the kernel size of each CNN, the number of neurons in convolution layer, the parameter of each GRU layer, the dropout rate, and the recurrent dropout rate. Finally, the prediction model, of which structure and hyperparameters are both in the optimal state, is obtained.

Step 4: **Model prediction.** The impact factors in the previously divided testing datasets are input into the trained model to predict the corresponding dam deformation.

Step 5: **Model evaluation.** Appropriate model evaluation indicators are selected and used to compare the accuracy, stability, and robustness of dam deformation prediction models based on MLP, CNN, LSTM, GRU, CNN–LSTM, and CNN–GRU.

Step 6: **Engineering application.** At this moment, each subcomponent trains its own optimal prediction model. The three simulated output variables should be renormalized to obtain true predicted values. In practice, the final predicted deformation result of dam is the sum of these three predicted components.

### 3. Case Study

#### 3.1. Study Area and Dataset

A parabolic variable-thickness double-curved arch dam located in China was considered as the research object, as shown in Figure 6. The crest elevation of the dam is 109 m, and the bottom elevation is 40.2 m. The dead water level of the reservoir is 64 m, and the corresponding storage capacity is 1.52 million m$^3$. This study selected dam deformation data as the analysis data source, which is mainly affected by the combined influence of the reservoir water level and temperature. For the arch dam, studying the deformation law for a reasonable prediction is the primary objective during the service of the arch dam.

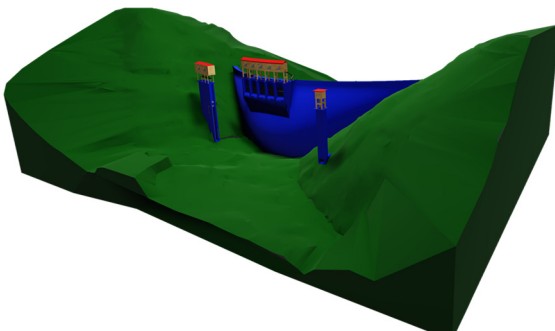

**Figure 6.** Panorama of the studied dam project.

Since the first deformation of the dam in 2009, four monitoring points have been configured on top of the dam, namely G0-HY, G1-HY, G2-HY, and G3-HY, which are measured by the Global Navigation Satellite Systems (GNSS). All the points contain complete records of deformation data during the past years, and the absolute radial displacement is selected as the object of analysis and prediction in this study. Figure 7 presents the absolute radial displacement data of the reservoirs, where the data for the arch dam were collected during the period from June 2009 to December 2019. Through observation, we find that the overall deformation data process line of the dam fluctuates regularly and is relatively stable, with, mostly, no evident saltation. Moreover, the deformation at the measuring points in the middle of the dam is greater than that at the measuring points located at both ends of the dam, and all the measuring points have a trend of convergence and periodicity. In this case, data from the initial 60% and 30% were used for training, respectively. The other data were used for testing. All the modules were implemented using Python.

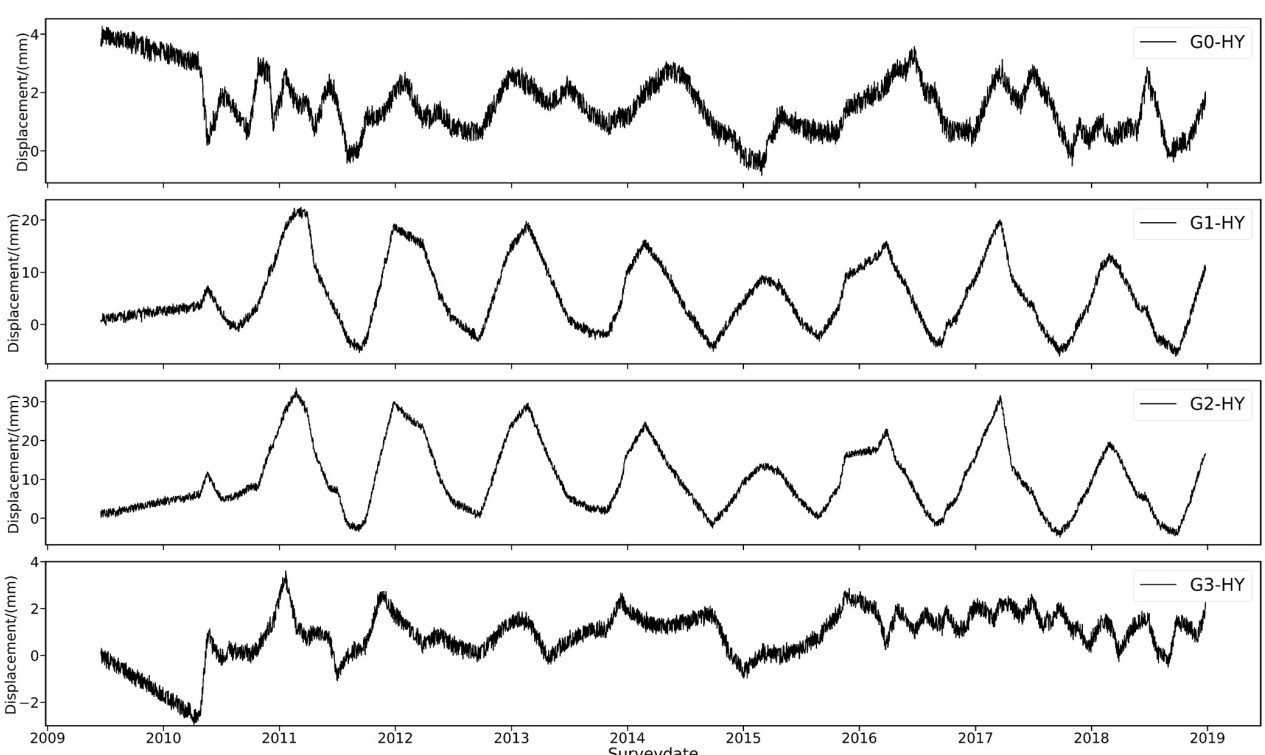

**Figure 7.** Radial displacement changes at each measuring point.

*3.2. Decomposition Results*

From the perspective of dam deformation characteristics, the setting of the sequence decomposition frequency is critical for subsequent model construction, and the dam body displacement changes annually. Therefore, the sequence decomposition frequency of the STL method is set to 365, and then the observed radial displacement data are decomposed into three components. The decomposition results show that all the components of the radial deformation time series are displayed as a cycle depending on the demand. Moreover, the trend component of G2-HY shows a gentle downward trend, whereas the trend component of G3-HY shows a much gentler growth. The residual component that shows a high degree of irregular complexity can be interpreted as the part where periodicity and the changing trend of the target quantity are eliminated. Because G0-HY and G3-HY have similar phenomena, G1-HY and G2-HY have similar phenomena; hence, Figure 8 shows only the representative components of the radial displacement data from G2-HY and G3-HY; the other measuring point results are not given. The components obtained by decomposition at each measuring point have evident differences, which shows the feasibility and variability of using STL to extract intrinsic information from the radial displacement time series.

*3.3. Input Variable Determination*

Previous research results have shown that the choice of input variables has a significant influence on the model prediction results; hence, the causes of various deformation components should be analyzed. The seasonal component is affected by the periodic changes in the hydrostatic load and seasonal temperature, the trend component is affected by the combined action of the dam material degradation and inherent rheological property, and the residual part is due to uncertain aging factors such as geological changes and structural damage [34–36]. Most of the previous statistical models were developed on the basis of the HST model, whereas the HST model does not include the hysteresis of the water level and air temperature relative to the dam deformation, and its simulation of the temperature effect through a simple harmonic function cannot accurately reflect the actual

temperature changes [37]. In this study, the HSTT model was used to simulate the observed dam deformation.

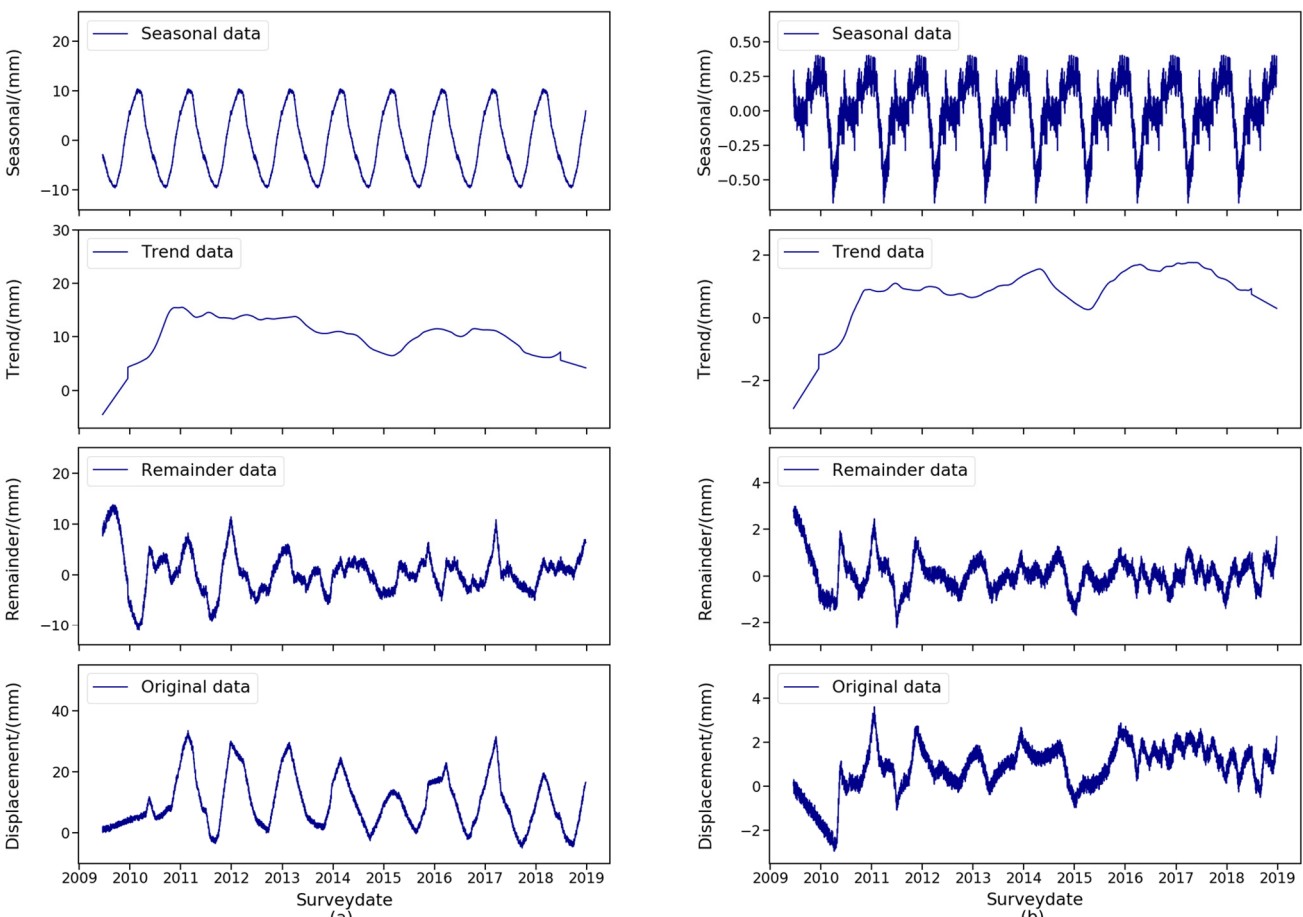

**Figure 8.** Decomposition results obtained using STL (**a**) G2-HY; (**b**) G3-HY.

The HSTT model is an improvement of the HST model, which adds an additional thermal term on the original basis to introduce a hysteresis description of the temperature effect. In the HSTT model, the reservoir level $H$ is used to describe the hydraulic effect $\delta(h)$. $\delta(h)$ can be considered as a linear polynomial with $H$, $H^2$, $H^3$, and $H^4$ terms. The thermal effect consists of a seasonal component $\delta(s)$ and an additional thermal term $\delta(T)$. The seasonal component $\delta(s)$ refers to the periodical variation in the dam deformation and can be modeled as $\sin\frac{2\pi t}{365}$, $\cos\frac{2\pi t}{365}$, $\sin\frac{4\pi t}{365}$, and $\cos\frac{4\pi t}{365}$. The additional thermal term $\delta(T)$ introduced to improve the simulation of the temperature effect is the difference between the fitted periodic function of the seasonal component $\delta(s)$ and the actual air temperature. The time-varying effect is the numerical response under the passage of time, which can be described with $t$ and $ln(t)$.

In short, according to the mathematical description of the HSTT model, the 11 influencing factors need to be used as the input variables of the models to predict the radial deformation of the dam.

### 3.4. Model Development

To illustrate the superiority of the hybrid model we proposed, it is compared with a total of nine models including multivariable linear regression (MLR), multilayer perceptron (MLP), 1D-CNN, LSTM, GRU, LSTM–SSA, GRU–SSA, CNN–LSTM–SSA, and CNN–GRU–SSA. This section provides detailed information about these models. The experimental configuration for this study was CPU Intel® Core i7-9750H and NVIDIA 1660Ti GPU. All algorithmic models in the framework were conducted in the python 3.6 environment, and the CNN, LSTM, and

GRU networks were implemented in the TensorFlow framework developed by the artificial intelligence team of Google. The final experimental results are listed in Tables 2 and 3.

**Table 2.** Basic information of model parameter tuning.

| Models | Hyperparameters | Search Range | Optimal Value | |
|---|---|---|---|---|
| | | | G2-HY | G3-HY |
| MLP | the number of neurons | (32, 128) | 60 | 64 |
| | dropout rate | [0.1, 0.2, 0.3, 0.4, 0.5] | 0.2 | 0.2 |
| CNN | the number of neurons | (64, 256) | 128, 70 | 62, 55 |
| | the size of kernel | [3, 5, 7] | 7 | 7 |
| | recurrent dropout rate | [0.3, 0.4, 0.5, 0.6, 0.7] | 0.4, 0.5 | 0.5, 0.5 |
| LSTM | the number of neurons | (64, 128) | 120, 70, 76 | 110, 56 |
| | recurrent dropout rate | [0.4, 0.5, 0.6, 0.7, 0.8] | 0.5, 0.5, 0.5 | 0.6, 0.6, 0.6 |
| GRU | the number of neurons | [64, 128] | 80, 64, 60 | 100, 78 |
| | recurrent dropout rate | [0.4, 0.5, 0.6, 0.7, 0.8] | 0.4, 0.5, 0.5 | 0.5, 0.5 |
| CNN–LSTM | the number of neurons for CNN | (128, 256) | 160 | 122 |
| | the size of kernel | [3, 5, 7] | 5 | 5 |
| | recurrent dropout rate for CNN | [0.2, 0.3, 0.4, 0.5, 0.6] | 0.5 | 0.5 |
| | the number of neurons for LSTM | (64, 128) | 117, 89 | 109, 88 |
| | recurrent dropout rate for LSTM | [0.2, 0.3, 0.4, 0.5, 0.6] | 0.4, 0.4 | 0.4, 0.5 |
| CNN–GRU | the number of neurons for CNN | (128, 256) | 146 | 115 |
| | the size of kernel | [3, 5, 7] | 7 | 7 |
| | recurrent dropout rate for CNN | [0.3, 0.4, 0.5, 0.6, 0.7] | 0.5 | 0.5 |
| | the number of neurons for GRU | (64, 521) | 108, 86 | 106, 77 |
| | recurrent dropout rate for GRU | [0.3, 0.4, 0.5, 0.6, 0.7] | 0.5, 0.5 | 0.5, 0.5 |

**Table 3.** The determined parameters of the proposed model.

| Hyperparameters | G2-HY | | | G3-HY | | |
|---|---|---|---|---|---|---|
| | Seasonality | Trend | Residual | Seasonality | Trend | Residual |
| the number of neurons for CNN | 128 | 156 | 222 | 68 | 108 | 188 |
| the size of kernel | 7 | 7 | 7 | 7 | 5 | 5 |
| recurrent dropout rate for CNN | 0.4 | 0.5 | 0.4 | 0.6 | 0.3 | 0.4 |
| the number of neurons for GRU | 146 | 133 | 196 | 128 | 88 | 214 |
| recurrent dropout rate for GRU | 0.5 | 0.5 | 0.3 | 0.5 | 0.4 | 0.5 |

(1)　MLP models

　　In addition to the input layer and output layer, the MLP model also added a hidden layer with a ReLU activation function to capture the nonlinear relationship of the target object. The SSA method was used to optimize the number of hidden units in order to obtain the prediction model with the minimum *RMSE* value. A dropout layer was introduced to regulate the participation of units during each training of the network, and to suppress the occurrence of overfitting.

(2)　CNN model

　　For the standard CNN model, which consists of a convolution and a pooling layer, the original radial dam displacement was selected as the target object. To read data sequentially, the max pooling layer was used, and a dropout layer was introduced to limit the network training process to suppress overfitting. The number of neurons and the size of the kernel in the convolution layer were determined by the SSA method. The output layer is a fully connected layer.

(3)    LSTM and GRU models

The deep LSTM and GRU models were used in this study, and the original radial dam displacement was selected as the target object. This stacked-layers mechanism made the output of the previous hidden layer from the model be the input of the next hidden layer. Moreover, adding a recurrent dropout layer limits the hyperparameters of the model from being too large, which would cause overfitting. The number of stacker-layers were set to two. The number of neurons and recurrent dropout rate were determined by the empirical method. The output layer is a fully connected layer.

(4)    LSTM–SSA and GRU–SSA

The purpose of establishing the LSTM–SSA model and the GRU–SSA model is to verify the importance of the optimal parameters. Then the SSA method found the appropriate value for the number of hidden neurons and the recurrent dropout rate. The model frameworks were the same as those discussed above.

(5)    CNN–LSTM–SSA and CNN–GRU–SSA models

The CNN layer was used for the convolution operation of input data to extract local features of the original time series. The LSTM and GRU layers were used to capture delay information of the time series. The computational parameters and model frameworks for CNN, LSTM, and GRU were the same as those discussed above.

*3.5. Evaluation Indicators*

This section discusses four evaluation indicators, namely the mean absolute error (*MAE*), mean squared error (*MSE*), root mean squared error (*RMSE*), and coefficient of determination ($R^2$), which are used to test the predictive abilities of different models. The formula of these four indexes are provided below.

$$MAE(y, \hat{y}) = \frac{1}{n} \sum_{i=1}^{n} |y_i - \hat{y}_i| \tag{17}$$

$$MSE(y, \hat{y}) = \frac{1}{n} \sum_{i=1}^{n} |y_i - \hat{y}_i|^2 \tag{18}$$

$$RMSE(y, \hat{y}) = \sqrt{\frac{1}{n} \sum_{i=1}^{n} |y_i - \hat{y}_i|^2} \tag{19}$$

$$R^2(y, \hat{y}) = 1 - \sum_{i=1}^{n} (y_i - \hat{y}_i)^2 / \sum_{i=1}^{n} (y_i - \overline{y}_i)^2 \tag{20}$$

where $\hat{y}$ is the predicted value of the *i*-th sample, $y_i$ is the corresponding true values, and $\overline{y} = \sum_{i=1}^{n} y_i / n$. These evaluation indicators can effectively observe the error between the predicted value and the measured value. It is worth mentioning that for all evaluation indicators except $R^2$, the smaller its value, the better performance of the forecasting model.

*3.6. Performance Comparison between Models with Different Time Steps*

Since the MLP architecture accepts 2D tensors and does not require time series reconstruction, while the CNN, LSTM, and GRU architectures accept 3D tensors, the time step needs to be optimized. Figure 9 shows the average MAE variation in the simulation results of the CNN, LSTM, GRU, CNN–LSTM, and CNN–GRU models with different time steps. Considering the influence of the sequence of data input on the simulation results, a five-fold cross-validation is used for the model. Therefore, the training and test sets are divided differently each time, which leads to differences in the results of the five iterations for a given time-step combination. In Figure 10, the maximum and minimum values of MAE are, respectively, used as the upper and lower boundaries of the band.

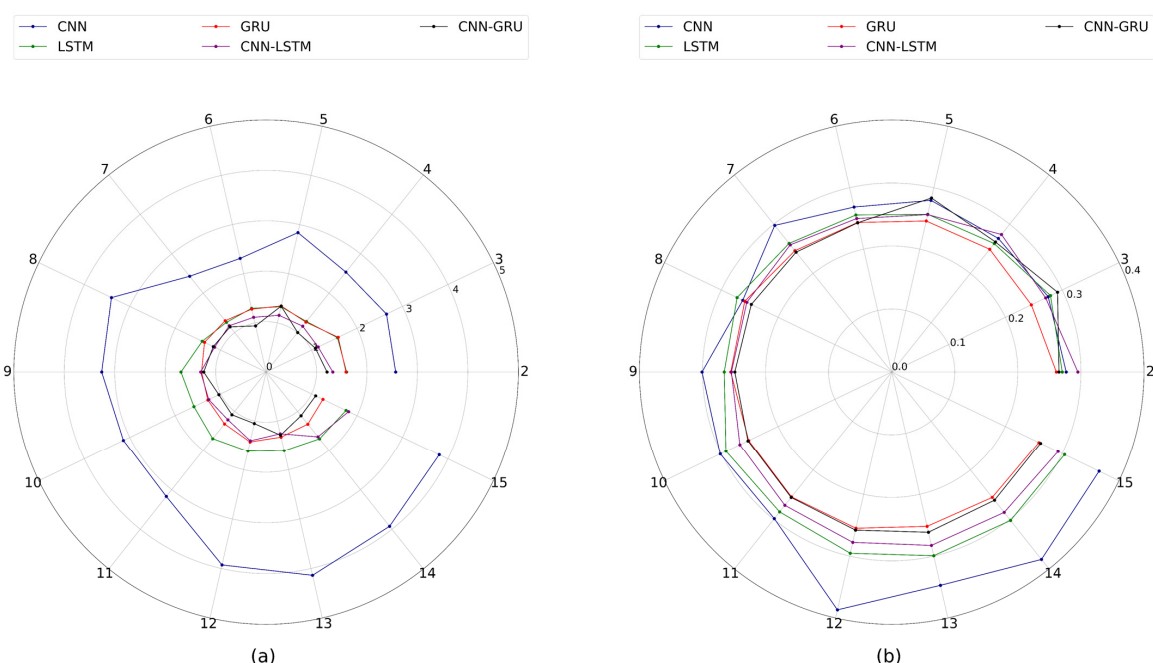

**Figure 9.** Average MAE values for (**a**) G2-HY and (**b**) G3-HY from CNN, LSTM, GRU, CNN–LSTM, and CNN–GRU models in five repeated simulations with different time steps.

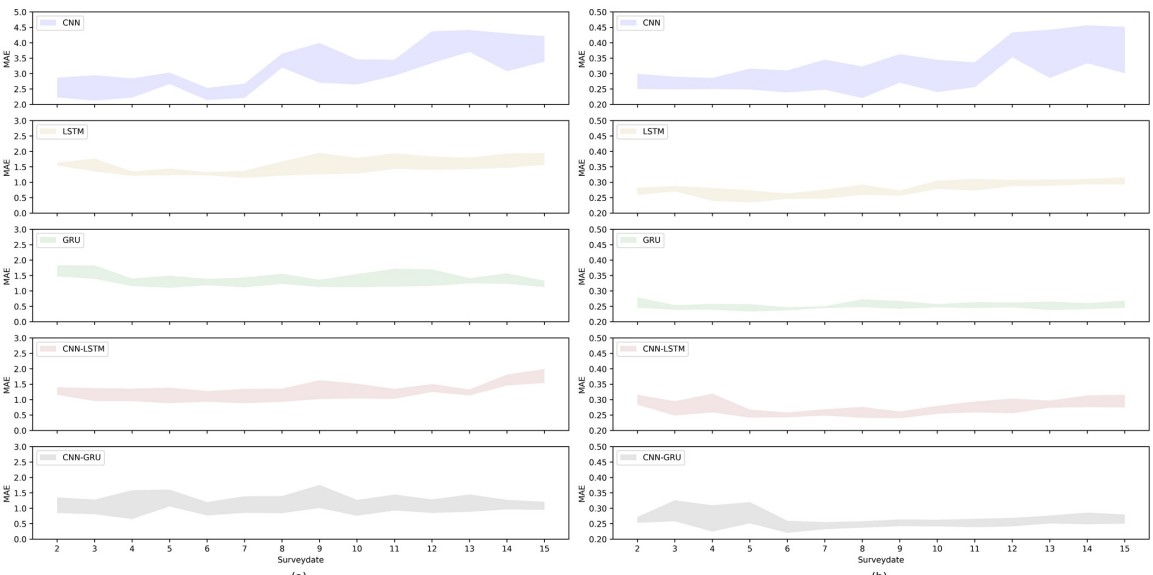

**Figure 10.** Maximum and minimum MAE values for (**a**) G2-HY and (**b**) G3-HY from CNN, LSTM, GRU, CNN–LSTM, and CNN–GRU models in five repeated simulations with different time steps.

The results show that if under a fixed prediction time, the MAE of the CNN models initially decreases with the increasing time step and then increases. To accurately predict the future dam deformation, the model should be inputted with time series data containing sufficient, complete information pertaining to dam deformation events. The length of the input time series is essentially the value of the time step. For CNN models with a prediction time of one day, when the time step is from one to six days, the accuracy of the dam deformation prediction increases with the increasing time step. However, when the time step exceeds six days, the input time series has redundant information, which will become a negative factor for dam deformation prediction. Therefore, when using the CNN model to predict dam deformation, the optimal time step required by the model should

be determined first, and the optimal value may change because data are collected from different dams.

On the other hand, the results show that it is not necessary to choose the optimal time step for the LSTM, GRU, CNN–LSTM, and CNN–GRU models. As shown in Figure 9, although the CNN–LSTM and CNN–GRU models are less accurate when the time step is small and when the MAE fluctuates, the overall MAE value decreases with the increasing time step for these four models and remains nearly stable when the time steps are large. This indicates that the need to eliminate redundant information in the LSTM, GRU, CNN–LSTM, and CNN–GRU models is not as important as in the CNN model. For these models, a relatively large time step can be selected in practical applications to allow the training dataset to introduce a sufficient time series, while the introduction of unnecessary information will not negatively affect the prediction accuracy.

Another phenomenon can be observed in Figure 10: When the time step is optimal, the MAE value band of the CNN is wider than those of the other models, indicating that the uncertainty in the prediction results of the CNN model is greater than those of the LSTM, GRU, CNN–LSTM, and CNN–GRU models, whereas the LSTM, GRU, CNN–LSTM, and CNN–GRU models have less uncertainty in practical dam deformation prediction, and repeated simulations may yield more consistent predictions.

The experimental results show that six days is the optimal time step for the five models. We use it to discuss the short-term, long-term, and abnormal deformation events in the case study dataset.

### 3.7. Simulation Results

Taking monitoring points G2-HY and G3-HY as representative objects, the other prediction results show similar phenomena and are not given herein. Based on the descriptions above, the original radial displacement–decomposed components were modeled using different models.

### 3.7.1. Short-Term Prediction Performance Evaluation

A measurement index is used to evaluate the prediction abilities of the nine models, as shown in Table 4. Figure 11 shows the residual boxplots of the proposed model and other conventional models for the two monitoring points. From Table 4 and Figure 11, we find that the hyperparameters of the architecture influence the prediction accuracy of the LSTM and GRU model. For instance, the MSE value of the LSTM–SSA is reduced by approximately 30.1% for G3-HY compared with the LSTM. Compared with the GRU, the GRU–SSA prediction results show a reduction of 10.9% in terms of the MAE for G3-HY. Combined with the boxplot in Figure 11c, the median values of the outliers of the model after hyperparameter optimization are lower than those of the model without optimization, indicating that their predictive performance is better and more stable, which can improve the model prediction performance.

The graphs on the right side of Figure 12 show a performance comparison between the models (i.e., LSTM–SSA, GRU–SSA, CNN–LSTM–SSA, CNN–GRU–SSA, and the proposed model) for short-term prediction. The models show good fitting results on the training dataset, indicating that each model can fit the dam deformation time series. It can be seen from Figure 12j that the prediction performance of the proposed model is outstanding among the nine models, and the predictive displacements are basically consistent with the observed data. Moreover, the LSTM–SSA and GRU–SSA models have prediction errors for the deformation monitoring data having crests and troughs. However, owing to the introduction of the CNN model, the prediction accuracy of the model for crests and troughs is improved. The feature extraction potentials of the CNNs are demonstrated in this case. Thus, the hybrid model reduces the computational pressure while improving the performance of extreme value prediction, so that the prediction model is more suitable for practical problems.

**Table 4.** Results of different models for an arch dam with 60% of the data used for training.

| Models | MAE | | MSE | | RMSE | | $R^2$ | |
|---|---|---|---|---|---|---|---|---|
| | G2-HY | G3-HY | G2-HY | G3-HY | G2-HY | G3-HY | G2-HY | G3-HY |
| Hybrid model | 0.7765 | 0.1623 | 0.9254 | 0.0382 | 0.9620 | 0.1954 | 0.9877 | 0.9327 |
| MLP | 2.0126 | 0.2791 | 6.6995 | 0.1190 | 2.588 | 0.3450 | 0.8719 | 0.7449 |
| CNN | 2.3113 | 0.2685 | 7.1580 | 0.1106 | 2.6754 | 0.3326 | 0.8686 | 0.7508 |
| LSTM | 1.8344 | 0.2961 | 6.4288 | 0.1379 | 2.5355 | 0.3713 | 0.8756 | 0.7235 |
| GRU | 1.7991 | 0.2735 | 5.1987 | 0.1146 | 2.2801 | 0.3385 | 0.8850 | 0.7420 |
| LSTM–SSA | 1.2957 | 0.2556 | 3.1196 | 0.0964 | 1.8492 | 0.3105 | 0.9302 | 0.7838 |
| GRU–SSA | 1.2801 | 0.2434 | 3.0687 | 0.0859 | 1.8079 | 0.2931 | 0.9340 | 0.7979 |
| CNN–LSTM–SSA | 1.1129 | 0.2498 | 2.1640 | 0.0906 | 1.4710 | 0.3009 | 0.9552 | 0.7981 |
| CNN–GRU–SSA | 0.9384 | 0.2429 | 1.4630 | 0.0873 | 1.2095 | 0.2955 | 0.9752 | 0.8260 |

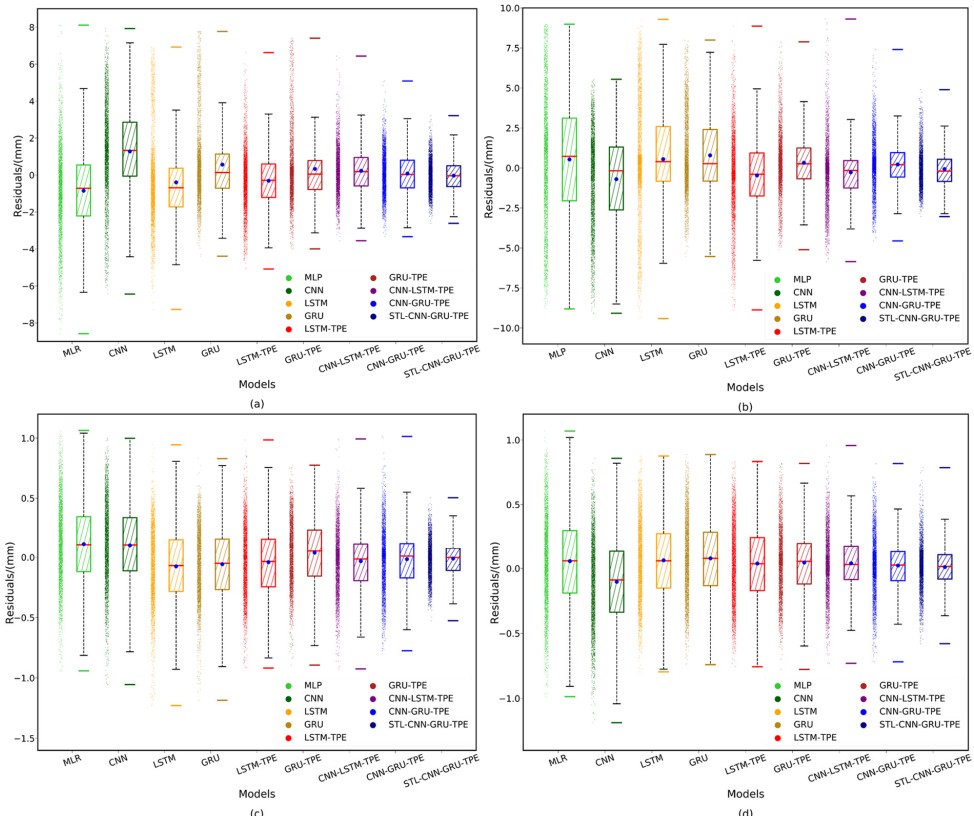

**Figure 11.** Boxplots of residuals from the hybrid model and the benchmark models: (**a**) 60% of the data of G2-HY for training; (**b**) 30% of the data of G2-HY for training; (**c**) 60% of the data of G3-HY for training; (**d**) 30% of the data of G3-HY for training.

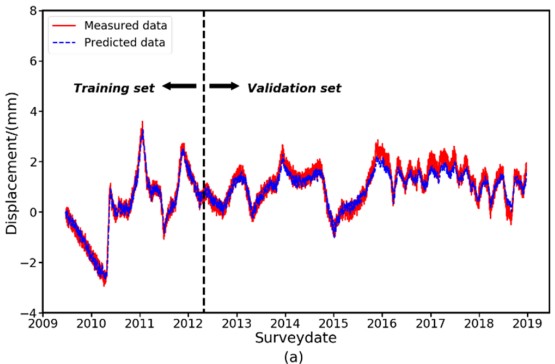 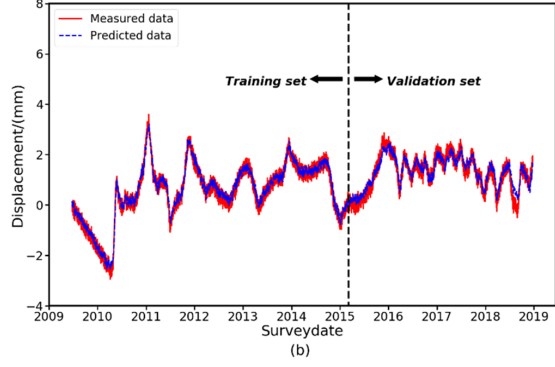

**Figure 12.** *Cont.*

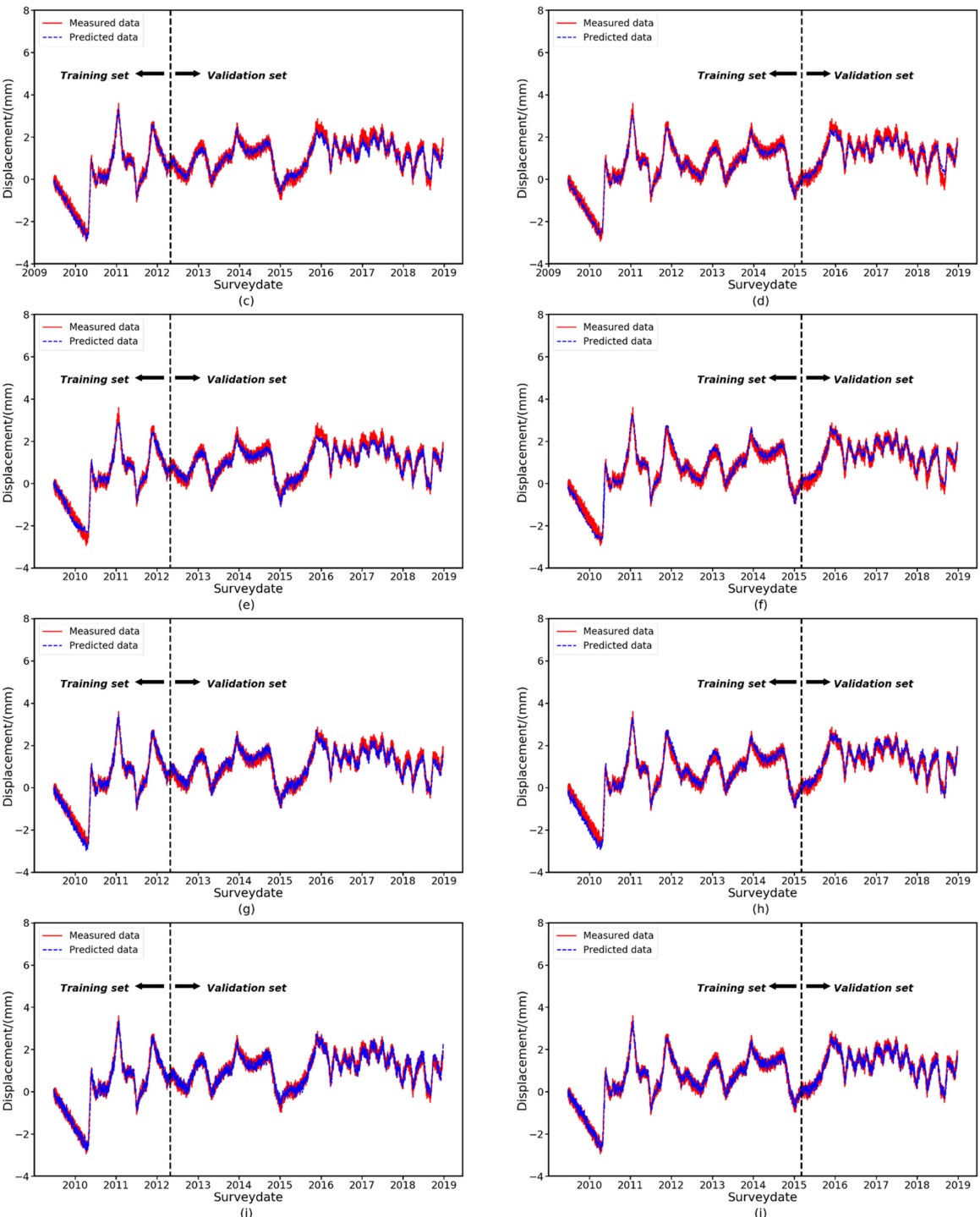

**Figure 12.** Long-term and short-term predictions for G3-HY through the different models: (**a**) LSTM-SSA with 30% data for training; (**b**) LSTM-SSA with 60% data for training; (**c**) GRU-SSA with 30% data for training; (**d**) GRU-SSA with 60% data for training; (**e**) CNN-LSTM-SSA with 30% data for training; (**f**) CNN-LSTM-SSA with 60% data for train-ing; (**g**) CNN-GRU-SSA with 30% data for training; (**h**) CNN-GRU-SSA with 60% data for training; (**i**) The hybrid model with 30% data for training; (**j**) The hybrid model with 60% data for training.

### 3.7.2. Long-Term Prediction Performance Evaluation

Most dams have less monitoring data. Therefore, less training data should be used to verify the stability of the prediction model for dam deformation. Two cases with different lengths of the training dataset were considered. In the first case, the monitoring dataset

was divided into 60% for training and 40% for testing. In the second case, only 30% of the data were used for training and 70% of the data for testing. The prediction results of the nine models are evaluated using measurement indicators, as shown in Table 2. Typically, the hybrid model outperforms the single-algorithm models (CNN, LSTM, and GRU). For example, compared with the CNN–GRU–SSA model, the proposed model reduces the *RMSE* by 20.6% for G3-HY. Compared with the CNN–LSTM–SSA model, the prediction accuracy of the proposed model improves significantly in the aspect of $R^2$ promotion of up to 14.2% for G3-HY. Combined with the boxplot, as shown in Figure 11d, the single-algorithm models have some outliers, whereas the proposed model is within the 1.5 interquartile range, and there are only a few mild outliers, indicating that the hybrid model is more suitable for long-term prediction than the single-algorithm models.

The five better models are selected here for comparison, as shown on the left side of Figure 12. The prediction accuracy of the proposed hybrid model reveals that using a shorter training dataset yields results similar to those using a longer training dataset. When the training data are reduced, the LSTM is more sensitive than the GRU, which shows that the prediction accuracy gradually decreases in the later stage, indicating that the GRU outperforms the LSTM in some tasks. Therefore, through an analysis of the long-term prediction ability of the model, it is proved that the effective combination of the method is beneficial to improve the generalization ability of the predictive model.

### 3.7.3. Evaluation of Abnormal Time Series Prediction Performance

There was an abrupt variation in the displacement during the period ranging from 27 April 2010 to 11 July 2010 for G2-HY, as shown in Figure 13. This abnormal phenomenon is usually caused by environmental factors or material properties. The prediction performances of the nine models are evaluated using the measurement indices, as shown in Tables 4 and 5. The proposed model significantly outperforms the other models on all the indicators, regardless of the proportion of data used for training (60% or 30%). For instance, the MAE value of the hybrid model is reduced by approximately 67.9% during the testing phase compared with that of the CNN model. The improvements in the MSE and $R^2$ are as high as 52.3% and 8.1% compared with the CNN–GRU–SSA model when in the testing phase, respectively. Combined with the boxplot, as shown in Figure 11a,b, extreme outliers exist in all the other models for the two monitoring points except in the case of the proposed model, while only slight outliers exist in the proposed model with a relatively concentrated residual distribution. Thus, the mutation resistance and robustness of the proposed model are proven.

**Table 5.** Results of different models for an arch dam with 30% of the data used for training.

| Models | *MAE* | | *MSE* | | *RMSE* | | $R^2$ | |
|---|---|---|---|---|---|---|---|---|
| | **G2-HY** | **G3-HY** | **G2-HY** | **G3-HY** | **G2-HY** | **G3-HY** | **G2-HY** | **G3-HY** |
| Hybrid model | 0.8340 | 0.1926 | 1.1624 | 0.0562 | 1.0782 | 0.2370 | 0.9842 | 0.9055 |
| MLP | 3.2137 | 0.3015 | 15.5721 | 0.1383 | 3.9461 | 0.3720 | 0.7377 | 0.6385 |
| CNN | 2.5986 | 0.2872 | 10.5526 | 0.1278 | 3.2484 | 0.3575 | 0.7958 | 0.6483 |
| LSTM | 2.7929 | 0.2520 | 7.0410 | 0.0953 | 2.8101 | 0.3087 | 0.8177 | 0.6845 |
| GRU | 1.9728 | 0.2536 | 6.6281 | 0.0964 | 2.5745 | 0.3105 | 0.8519 | 0.7199 |
| LSTM–SSA | 2.3448 | 0.2367 | 5.8182 | 0.0829 | 2.4030 | 0.2880 | 0.8736 | 0.7744 |
| GRU–SSA | 1.7885 | 0.2379 | 5.3936 | 0.0834 | 2.3224 | 0.2888 | 0.8980 | 0.7747 |
| CNN–LSTM–SSA | 1.8223 | 0.2345 | 5.5960 | 0.0806 | 2.3656 | 0.2839 | 0.8994 | 0.7930 |
| CNN–GRU–SSA | 1.7745 | 0.2378 | 5.1085 | 0.0847 | 2.2602 | 0.2910 | 0.9051 | 0.8073 |

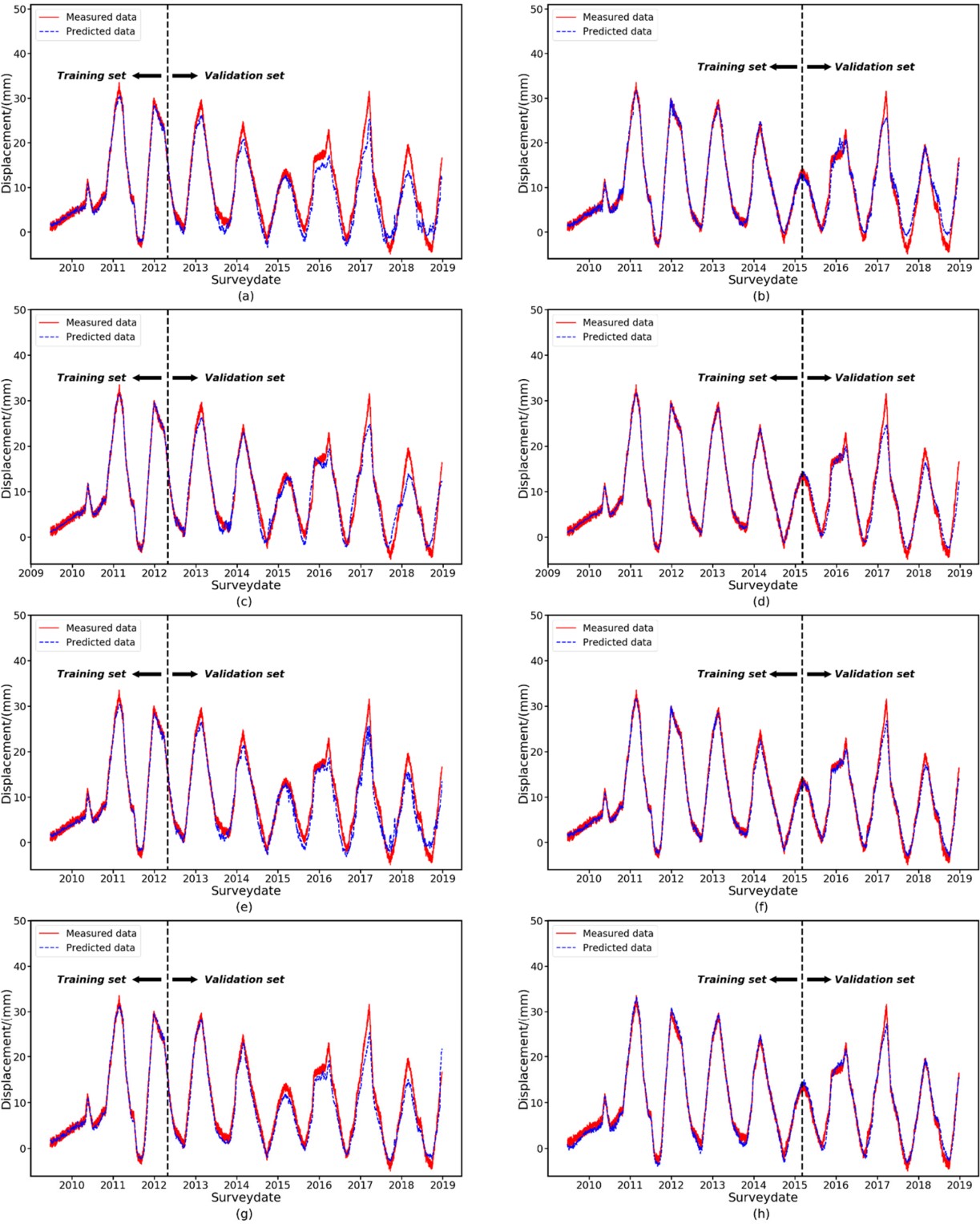

**Figure 13.** *Cont.*

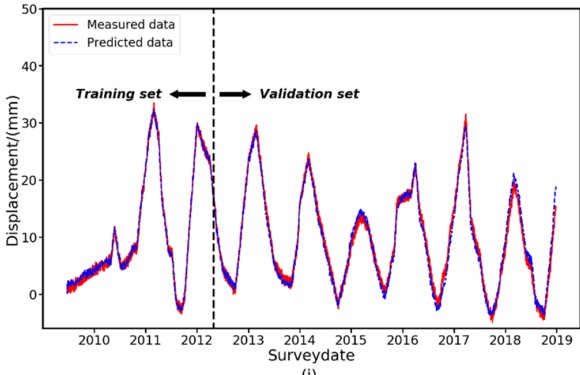
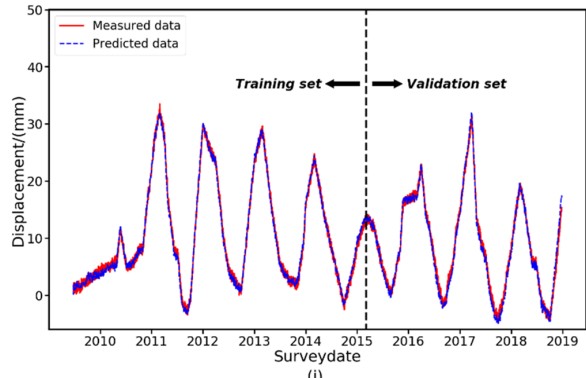

**Figure 13.** Long-term and short-term predictions for G2-HY through the different models: (**a**) LSTM-SSA with 30% data for training; (**b**) LSTM-SSA with 60% data for training; (**c**) GRU-SSA with 30% data for training; (**d**) GRU-SSA with 60% data for training; (**e**) CNN-LSTM-SSA with 30% data for training; (**f**) CNN-LSTM-SSA with 60% data for train-ing; (**g**) CNN-GRU-SSA with 30% data for training; (**h**) CNN-GRU-SSA with 60% data for training; (**i**) The hybrid model with 30% data for training; (**j**) The hybrid model with 60% data for training.

The five better models are selected here for comparison, as shown in Figure 13. All the prediction models can fit the changes in the abnormal phenomena when 60% of the data are used for training. When 30% of the data are used for training, the other models will be polluted training dataset; only the proposed hybrid model performs well. Therefore, even if the monitoring time series has abnormal data, the proposed hybrid model still has good generalization performance, indicating that it is a promising dam deformation prediction model.

### 3.8. Discussion of Results

The simulation results showed that there are differences between the MLP, CNN, LSTM, and GRU models, which indicates the importance of method selection when constructing prediction models. For the dam deformation problem, the prediction accuracies of the GRU and LSTM are similar and remain stable when the time step is large, which avoids the trouble of optimizing the time step. However, the GRU model may be preferred because of its simple structure, fewer parameters, and less time consumption in model training. When using gradient descent for deep learning model optimization, there are shortcomings such as overfitting and the choice of calculation parameters [38].

In this study, the SSA method uses the adaptive scores in previous iterations to learn and select the best set of hyperparameters. To avoid overfitting, the use of recurrent dropout techniques can limit the participation of hidden units in the training process of the deep learning model. To a certain extent, it not only overcomes the existing shortcomings of the deep learning models, but also provides relatively well-predicted results. However, the deformation is affected by many external factors and the dam materials used, resulting in complexity, uncertainty, diversity, and time variation in the deformation series [5]. The noise, nonstationarity, and heteroscedastic structure of the dam deformation time series often affect the ability of models to accurately and comprehensively reflect the formation factors and changing laws of dam deformation when a single-prediction model is used, indicating that the predicted results are affected by the high volatility of the data [19]. To avoid this problem, a decomposition method is used to identify sequence features and accurately separate the sequences into multiple components for prediction, thereby significantly reducing modeling difficulty. In addition, the CNN can capture sequence characteristics through a convolution kernel. The GRU can learn these characteristics through forward and backward dependencies. Although the hybrid model has poor prediction accuracy stability when the time step is small, the prediction accuracy tends to stabilize as the time

step increases. Therefore, the STL–CNN–GRU hybrid model outperforms the standard deep learning methods.

In the following, we analyze the possible reasons why the proposed model is better than the other models. First, the STL decomposes the observed deformation time series into three components, which is convenient for effectively identifying the influence of various factors. Second, the CNN–GRU model is used to establish a complex mapping weight matrix of the input variables and output targets of each component. The CNN is used for convolution operation of the input data to extract the characteristics of the time series data. To learn the peak characteristics, the GRU is used to store the delay information in these time series. To effectively improve the generalization ability of CNN–GRUs, the SSA method is used to specify the optimal hyperparameters of the model. Finally, the simulation output results of all the CNN–GRU models are summed to obtain a time-sensitive final prediction result, which makes the model have the generalization ability for time series with different lengths and abnormal data to a certain extent. In summary, incorporating data decomposition and automatic hyperparameter optimization into the prediction model reduces modeling difficulty on the one hand, and improves the accuracy of the prediction results on the other hand, providing valuable technical references for dam deformation prediction.

## 4. Conclusions and Future Work

In recent decades, methods that can enhance the accuracy of dam deformation prediction have been widely studied in the dam safety monitoring field. To enrich the theory of dam safety monitoring, we proposed a hybrid prediction model according to the factors influencing the deformation of arch dams. First, the original dam deformation series was decomposed into season, trend, and remainder components through the STL method. Second, an CNN–GRU model optimized using the SSA was applied to predict the three components. Finally, the outputs of all component prediction models were summed and used as the final predicted value of the dam deformation. The performance of the proposed model was tested using a parabolic variable-thickness double-curved arch dam located in China as a practical example. The results showed that the hybrid model has better forecasting effects on time series of different lengths and abnormal data compared with several existing models with statistical indicators. Therefore, the good prediction ability makes it an effective tool for dam safety monitoring tasks. The main conclusions of this work are summarized as follows:

(1) A time series decomposition method was used to divide the components of the deformation series to strengthen the data characteristics, thus eliminating the volatility of the dam deformation monitoring time series to a certain extent and reducing modeling difficulty.

(2) The prediction accuracies of the GRU and LSTM models are largely identical for short-term deformation prediction and abnormal data series prediction. They were not affected by the redundant information introduced by a larger time step, and repeated simulations could yield more consistent prediction results with less uncertainty. The GRU exhibited a better prediction performance than the LSTM for long-term deformation prediction. In addition, the GRU may be the preferred method owing to its simpler structure, fewer parameters, and less training time consumption.

(3) The SSA method fully explored the nonlinear mapping potential of the CNN and GRU by selecting the optimal parameter set; thus, the hybrid model exhibited better accuracy, stability, and robustness than the existing model in practical application.

(4) The multi-model fusion strategy makes full use of the feature extraction potential of the CNN and the temporal feature representation of the GRU. Distributed modeling helps realize a targeted modeling of the deformation time series with different features to a certain extent and extends the scope of the model application.

Although the feasibility of the dam deformation prediction model proposed in this study was verified using a practical case, there are still some research directions that should

be carried out in the future. The first is the consideration of the impact of geospatial location factors between the monitoring points on dam deformation. The second is to solve the problem that the original dataset is not enough to build a predictive model for improving the engineering application value of the predicted model. The third is to determine early warning guidelines for dam safety to meet the requirements of life management. It is necessary to make an intensive study of dam safety monitoring to provide support for dam operation and management.

**Author Contributions:** C.L. and K.W. conceived and designed the experiments and performed the modeling. Y.L. and Q.H. provided relevant technical support. T.Z. and Y.S. analyzed the data and reviewed and edited the paper. All authors have read and agreed to the published version of the manuscript.

**Funding:** This work was supported by the National Natural Science Foundation of China (Grant No. 52109118), the Young Scientist Program of Fujian Province Natural Science Foundation (Grant No. 2020J05108), and Talent Introduction Scientific Start-up Foundation of Fuzhou University (Grant No. 510890).

**Conflicts of Interest:** The authors declare no conflict of interest.

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
