# Peer review of "Time Series Prediction of Dam Deformation Using a Hybrid STL–CNN–GRU Model Based on Sparrow Search Algorithm Optimization"

_applsci, doi:10.3390/app122311951_

Round 1

Reviewer 1 Report

In general, this paper shows the research gap and contributes the use of hybrid STL–CNN–GRU model-based sparrow search algorithm optimization in predicting of dam deformation. 

However, the authors should respond the following concerns:

1.     Abstract. Please add how much strong robustness to abnormal sequences than other conventional models were performed to make it clear the statement in Line 25.

2.     Introduction. Please ensure the formatting of durability1 (Line 37) and dam2 (Line 41). Is it reference number? please follow [1], and so on.

3.     Methods. The writing of symbols such as Yt (Line 178), St, Tt, and Rt (Line 179) not formal and seems superscript. 

4.     There was an Error! Reference source not found.. (Line 194). Similar case for Line 209, 226, 240, 250, 274, 322, 383, 404

5.      Line 373, the authors stated Fig.1, however, in previous stated Figure 1. Not consistent. 

6.     Line 429, the authors wrote not proper formula.

7.     Fig.2 and Figure 10 (between Line 525-526) were not clear. It is suggested to enhance the quality of both figures.

8.     The author should add some explanations in giving statement in Line 523-525 due to Fig.2 and Figure 10 (between Line 525-526) shown the phenomena and contribution to this study

Reviewer 2 Report

1.       Why is there a need to optimize the deep network as they are self-sufficient of themselves to adopt any situation of complexity?

2.       Why specifically sparrow search algorithm is selected for optimization? Is there any logical reason?

3.       In the Abstract section, some quantitative figures in terms of performance improvement comparative to the naive method should be stated as a brief summary. e.g. % of improvement over the other methods, how much less time etc.

4.       Authors should add a paragraph into the introduction section,. They should write "The main contributions of this paper are: (i) ….. (ii) ……. and (iii) ……" to highlight the key works. By this way, authors should provide a stronger motivation clearly and explain the originality of the paper. Please, provide a paragraph with three to five clear positive impacts of the proposed algorithm.

5.       The research gaps and contributions should be clearly summarized in introduction section.  The authors just described the related works that the researchers have done, but they did not evaluate the advantages and disadvantages of the related works. Please evaluate that how their study is different from others in the related work section? What do they have where others do not? Why they are better or how?  The authors should discuss the literature review more deeply and clearly. Try to make the paragraphs in the introduction section more comprehensive, it is very short. Authoritative synthesis assessing the current state-of-the-art is absent. The current introduction is simple and misses many contents related to the problem formulation. The authors are supposed to focus on the main topic of the study and present a Literature Review in the form of tables in order to make research gaps and innovations easy to detect. There is not a clear categorization of related works.

6.       What are the other possible methodologies that can be used to achieve your objective in relation in this work?

7.       What are the limitation(s) of the methodology adopted in this work? Discuss the limitations of your study. These limitations can be organized around simple distinctions of the choices you made in your study regarding who, what, where, when, why, and how.

8.       “Error! Reference source not found.” in the manuscript must be corrected. In this current form, many of the citations cannot be tracked and checked.

9.       All references should be checked. All variables should be written in italic. Some mathematical notations are not rigorous enough to correctly understand the contents of the paper. The authors are requested to recheck all the definition of variables and further clarify these equations.

10.   Figures should be polished. Some of them does not have axis names.

11.   It is not clear if experimental results were obtained under the same experimental conditions. Are the simulations performed in the same situations? How do you guarantee a fair comparison?  Add further details on how simulations were conducted. Similarly, resource and system characteristics could be added to Tables for clarity. The paper lacks the running environment, including hardware and software details. The analysis and configurations of experiments should be presented in detail for reproducibility. It is convenient for other researchers to redo your experiments and this makes your work easy acceptance. A table with parameter setting for experimental results and analysis should be included in order to clearly describe them.

12.   All of the values for the parameters of all algorithms selected for comparison are not given.

13.   Encoding type of the sparrow search algorithm for the focused problem should be detailed. It can also be shown as figure. Decision variables, their types, their boundaries, etc. should be shown. If normalization for the variables has been performed for the algorithms, this should be written within the text. Fitness function used for intelligent optimization method is also not clear.

14.   How the constraints are coped with is not clear. Constraint handling method for the sparrow search algorithm is not described.

Round 2

Reviewer 2 Report

The current version of the paper presents an expressive improvement as compared to the previous one. The authors provided acceptable answers to all questions and no more issues were detected in the current manuscript. Therefore, this reviewer recommends the acceptance of the paper in its current form.